# Asymmetric LSH (ALSH) for Sublinear Time Maximum Inner Product Search (MIPS)

**Anshumali Shrivastava**
Department of Computer Science
Computing and Information Science
Cornell University
Ithaca, NY 14853, USA
anshu@cs.cornell.edu

**Ping Li**
Department of Statistics and Biostatistics
Department of Computer Science
Rutgers University
Piscataway, NJ 08854, USA
pingli@stat.rutgers.edu

## Abstract

We present the first provably sublinear time hashing algorithm for approximate *Maximum Inner Product Search* (MIPS). Searching with (un-normalized) inner product as the underlying similarity measure is a known difficult problem and finding hashing schemes for MIPS was considered hard. While the existing Locality Sensitive Hashing (LSH) framework is insufficient for solving MIPS, in this paper we extend the LSH framework to allow asymmetric hashing schemes. Our proposal is based on a key observation that the problem of finding maximum inner products, after independent asymmetric transformations, can be converted into the problem of approximate near neighbor search in classical settings. This key observation makes efficient sublinear hashing scheme for MIPS possible. Under the extended asymmetric LSH (ALSH) framework, this paper provides an example of explicit construction of provably fast hashing scheme for MIPS. Our proposed algorithm is simple and easy to implement. The proposed hashing scheme leads to significant computational savings over the two popular conventional LSH schemes: (i) Sign Random Projection (SRP) and (ii) hashing based on $p$-stable distributions for $L_2$ norm (L2LSH), in the collaborative filtering task of item recommendations on Netflix and Movielens (10M) datasets.

## 1 Introduction and Motivation

The focus of this paper is on the problem of *Maximum Inner Product Search (MIPS)*. In this problem, we are given a giant data vector collection $\mathcal{S}$ of size $N$, where $\mathcal{S} \subset \mathbb{R}^D$, and a given query point $q \in \mathbb{R}^D$. We are interested in searching for $p \in \mathcal{S}$ which maximizes (or approximately maximizes) the **inner product** $q^T p$. Formally, we are interested in efficiently computing

$$p = \arg\max_{x \in \mathcal{S}} \quad q^T x \tag{1}$$

The MIPS problem is related to *near neighbor search (NNS)*, which instead requires computing

$$p = \arg\min_{x \in \mathcal{S}} \|q - x\|_2^2 = \arg\min_{x \in \mathcal{S}} (\|x\|_2^2 - 2q^T x) \tag{2}$$

These two problems are equivalent if the norm of every element $x \in \mathcal{S}$ is constant. Note that the value of the norm $\|q\|_2$ has no effect as it is a constant and does not change the identity of $\arg\max$ or $\arg\min$. There are many scenarios in which MIPS arises naturally at places where the norms of the elements in $\mathcal{S}$ have significant variations [13] and cannot be controlled, e.g., (i) recommender system, (ii) large-scale object detection with DPM, and (iii) multi-class label prediction.

**Recommender systems:** Recommender systems are often based on collaborative filtering which relies on past behavior of users, e.g., past purchases and ratings. Latent factor modeling based on matrix factorization [14] is a popular approach for solving collaborative filtering. In a typical matrix factorization model, a user $i$ is associated with a latent user characteristic vector $u_i$, and similarly, an item $j$ is associated with a latent item characteristic vector $v_j$. The rating $r_{i,j}$ of item $j$ by user $i$ is modeled as the **inner product** between the corresponding characteristic vectors.

In this setting, given a user $i$ and the corresponding learned latent vector $u_i$ finding the right item $j$, to recommend to this user, involves computing

$$j = \arg\max_{j'} \ r_{i,j'} = \arg\max_{j'} \ u_i^T v_{j'} \tag{3}$$

which is an instance of the standard MIPS problem. It should be noted that we do not have control over the norm of the learned vector, i.e., $\|v_j\|_2$, which often has a wide range in practice [13].

If there are $N$ items to recommend, solving (3) requires computing $N$ inner products. Recommendation systems are typically deployed in on-line application over web where the number $N$ is huge. A brute force linear scan over all items, for computing $\arg\max$, would be prohibitively expensive.

**Large-scale object detection with DPM:** Deformable Part Model (DPM) based representation of images is the state-of-the-art in object detection tasks [8]. In DPM model, firstly a set of part filters are learned from the training dataset. During detection, these learned filter activations over various patches of the test image are used to score the test image. The activation of a filter on an image patch is an inner product between them. Typically, the number of possible filters are large (e.g., millions) and so scoring the test image is costly. Recently, it was shown that scoring based only on filters with high activations performs well in practice [7]. Identifying those filters having high activations on a given image patch requires computing top inner products. Consequently, an efficient solution to the MIPS problem will benefit large scale object detections based on DPM.

**Multi-class (and/or multi-label) prediction:** The models for multi-class SVM (or logistic regression) learn a weight vector $w_i$ for each of the class label $i$. After the weights are learned, given a new test data vector $x_{test}$, predicting its class label is basically an MIPS problem:

$$y_{test} = \arg\max_{i \in \mathcal{L}} \ x_{test}^T \ w_i \tag{4}$$

where $\mathcal{L}$ is the set of possible class labels. Note that the norms of the vectors $\|w_i\|_2$ are not constant. The size, $|\mathcal{L}|$, of the set of class labels differs in applications. Classifying with large number of possible class labels is common in multi-label learning and fine grained object classification, for instance, prediction task with $|\mathcal{L}| = 100,000$ [7]. Computing such high-dimensional vector multiplications for predicting the class label of a single instance can be expensive in, e.g., user-facing applications.

## 1.1 The Need for Hashing Inner Products

Solving the MIPS problem can have significant practical impact. [19, 13] proposed solutions based on tree data structure combined with branch and bound space partitioning technique similar to k-d trees [9]. Later, the same method was generalized for general max kernel search [5], where the run-time guarantees, like other space partitioning methods, are heavily dependent on the dimensionality and the expansion constants. In fact, it is well-known that techniques based on space partitioning (such as k-d trees) suffer from the curse of dimensionality. For example, [24] showed that techniques based on space partitioning degrade to linear search, even for dimensions as small as 10 or 20.

Locality Sensitive Hashing (LSH) [12] based randomized techniques are common and successful in industrial practice for efficiently solving NNS (*near neighbor search*). Unlike space partitioning techniques, both the running time as well as the accuracy guarantee of LSH based NNS are in a way independent of the dimensionality of the data. This makes LSH suitable for large scale processing system dealing with ultra-high dimensional datasets which are common in modern applications. Furthermore, LSH based schemes are massively parallelizable, which makes them ideal for modern "Big" datasets. The prime focus of this paper will be on efficient hashing based algorithms for MIPS, which do not suffer from the curse of dimensionality.

## 1.2 Our Contributions

We develop *Asymmetric LSH (ALSH)*, an extended LSH scheme for efficiently solving the approximate MIPS problem. Finding hashing based algorithms for MIPS was considered hard [19, 13]. We formally show that, under the current framework of LSH, there cannot exist any LSH for solving MIPS. Despite this negative result, we show that it is possible to relax the current LSH framework to allow asymmetric hash functions which can efficiently solve MIPS. This generalization comes with no extra cost and the ALSH framework inherits all the theoretical guarantees of LSH.

Our construction of asymmetric LSH is based on an interesting fact that the original MIPS problem, after asymmetric transformations, reduces to the problem of approximate near neighbor search in

classical settings. Based on this key observation, we provide an example of explicit construction of asymmetric hash function, leading to the first provably sublinear query time hashing algorithm for approximate similarity search with (un-normalized) inner product as the similarity. The new ALSH framework is of independent theoretical interest. We report other explicit constructions in [22, 21].

We also provide experimental evaluations on the task of recommending top-ranked items with collaborative filtering, on Netflix and Movielens (10M) datasets. The evaluations not only support our theoretical findings but also quantify the obtained benefit of the proposed scheme, in a useful task.

## 2 Background

### 2.1 Locality Sensitive Hashing (LSH)

A commonly adopted formalism for approximate near-neighbor search is the following:

**Definition:** ($c$-Approximate Near Neighbor or $c$-NN) *Given a set of points in a $D$-dimensional space $\mathbb{R}^D$, and parameters $S_0 > 0$, $\delta > 0$, construct a data structure which, given any query point $q$, does the following with probability $1 - \delta$: if there exists an $S_0$-near neighbor of $q$ in $P$, it reports some $cS_0$-near neighbor of $q$ in $P$.*

In the definition, the $S_0$-near neighbor of point $q$ is a point $p$ with $Sim(q, p) \geq S_0$, where $Sim$ is the similarity of interest. Popular techniques for $c$-NN are often based on *Locality Sensitive Hashing* (LSH) [12], which is a family of functions with the nice property that more similar objects in the domain of these functions have a higher probability of colliding in the range space than less similar ones. In formal terms, consider $\mathcal{H}$ a family of hash functions mapping $\mathbb{R}^D$ to a set $\mathcal{I}$.

**Definition:** (Locality Sensitive Hashing (LSH)) *A family $\mathcal{H}$ is called $(S_0, cS_0, p_1, p_2)$-sensitive if, for any two point $x, y \in \mathbb{R}^D$, $h$ chosen uniformly from $\mathcal{H}$ satisfies the following:*

- *if $Sim(x, y) \geq S_0$ then $Pr_{\mathcal{H}}(h(x) = h(y)) \geq p_1$*
- *if $Sim(x, y) \leq cS_0$ then $Pr_{\mathcal{H}}(h(x) = h(y)) \leq p_2$*

For efficient approximate nearest neighbor search, $p_1 > p_2$ and $c < 1$ is needed.

**Fact 1 [12]:** Given a family of $(S_0, cS_0, p_1, p_2)$ -sensitive hash functions, one can construct a data structure for $c$-NN with $O(n^\rho \log n)$ query time and space $O(n^{1+\rho})$, where $\rho = \frac{\log p_1}{\log p_2} < 1$.

### 2.2 LSH for $L_2$ Distance (L2LSH)

[6] presented a novel LSH family for all $L_p$ ($p \in (0, 2]$) distances. In particular, when $p = 2$, this scheme provides an LSH family for $L_2$ distances. Formally, given a fixed (real) number $r$, we choose a random vector $a$ with each component generated from i.i.d. normal, i.e., $a_i \sim N(0, 1)$, and a scalar $b$ generated uniformly at random from $[0, r]$. The hash function is defined as:

$$h_{a,b}^{L2}(x) = \left\lfloor \frac{a^T x + b}{r} \right\rfloor \tag{5}$$

where $\lfloor \rfloor$ is the floor operation. The collision probability under this scheme can be shown to be

$$Pr(h_{a,b}^{L2}(x) = h_{a,b}^{L2}(y)) = F_r(d); \quad F_r(d) = 1 - 2\Phi(-r/d) - \frac{2}{\sqrt{2\pi}(r/d)}\left(1 - e^{-(r/d)^2/2}\right) \tag{6}$$

where $\Phi(x) = \int_{-\infty}^{x} \frac{1}{\sqrt{2\pi}} e^{-\frac{x^2}{2}} dx$ is the cumulative density function (cdf) of standard normal distribution and $d = \|x - y\|_2$ is the Euclidean distance between the vectors $x$ and $y$. This collision probability $F_r(d)$ is a monotonically decreasing function of the distance $d$ and hence $h_{a,b}^{L2}$ is an LSH for $L_2$ distances. This scheme is also the part of LSH package [1]. Here $r$ is a parameter. As argued previously, $\|x - y\|_2 = \sqrt{(\|x\|_2^2 + \|y\|_2^2 - 2x^T y)}$ is not monotonic in the inner product $x^T y$ unless the given data has a constant norm. Hence, $h_{a,b}^{L2}$ is not suitable for MIPS.

The recent work on *coding for random projections* [16] showed that L2LSH can be improved when the data are normalized for building large-scale linear classifiers as well as near neighbor search [17]. In particular, [17] showed that 1-bit coding (i.e., sign random projections (SRP) [10, 3]) or 2-bit coding are often better compared to using more bits. It is known that SRP is designed for retrieving with cosine similarity: $Sim(x, y) = \frac{x^T y}{\|x\|_2 \|y\|_2}$. Again, ordering under this similarity can be very different from the ordering of inner product and hence SRP is also unsuitable for solving MIPS.

# 3 Hashing for MIPS

## 3.1 A Negative Result

We first show that, under the current LSH framework, it is impossible to obtain a locality sensitive hashing scheme for MIPS. In [19, 13], the authors also argued that finding locality sensitive hashing for inner products could be hard, but to the best of our knowledge we have not seen a formal proof.

**Theorem 1** *There cannot exist any LSH family for MIPS.*

**Proof:** *Suppose there exists such hash function $h$. For un-normalized inner products the self similarity of a point $x$ with itself is $Sim(x,x) = x^T x = \|x\|_2^2$ and there may exist another points $y$, such that $Sim(x,y) = y^T x > \|x\|_2^2 + C$, for any constant $C$. Under any single randomized hash function $h$, the collision probability of the event $\{h(x) = h(x)\}$ is always 1. So if $h$ is an LSH for inner product then the event $\{h(x) = h(y)\}$ should have higher probability compared to the event $\{h(x) = h(x)\}$, since we can always choose $y$ with $Sim(x,y) = S_0 + \delta > S_0$ and $cS_0 > Sim(x,x)$ $\forall S_0$ and $c < 1$. This is not possible because the probability cannot be greater than 1. This completes the proof.* □

## 3.2 Our Proposal: Asymmetric LSH (ALSH)

The basic idea of LSH is probabilistic bucketing and it is more general than the requirement of having a single hash function $h$. The classical LSH algorithms use the same hash function $h$ for both the preprocessing step and the query step. One assigns buckets in the hash table to all the candidates $x \in S$ using $h$, then uses the same $h$ on the query $q$ to identify relevant buckets. The only requirement for the proof of Fact 1, to work is that the collision probability of the event $\{h(q) = h(x)\}$ increases with the similarity $Sim(q,x)$. The theory [11] behind LSH still works if we use hash function $h_1$ for preprocessing $x \in S$ and a different hash function $h_2$ for querying, as long as the probability of the event $\{h_2(q) = h_1(x)\}$ increases with $Sim(q,x)$, and there exist $p_1$ and $p_2$ with the required property. The traditional LSH definition does not allow this asymmetry but it is not a required condition in the proof. For this reason, we can relax the definition of $c$-NN without losing runtime guarantees. [20] used a related (asymmetric) idea for solving 3-way similarity search.

We first define a modified locality sensitive hashing in a form which will be useful later.

**Definition:** (*Asymmetric* Locality Sensitive Hashing (ALSH)) A family $\mathcal{H}$, along with the two vector functions $Q : \mathbb{R}^D \mapsto \mathbb{R}^{D'}$ (*Query Transformation*) and $P : \mathbb{R}^D \mapsto \mathbb{R}^{D'}$ (*Preprocessing Transformation*), is called $(S_0, cS_0, p_1, p_2)$-sensitive if, for a given $c$-NN instance with query $q$ and any $x$ in the collection $S$, the hash function $h$ chosen uniformly from $\mathcal{H}$ satisfies the following:

- if $Sim(q,x) \geq S_0$ then $Pr_{\mathcal{H}}(h(Q(q))) = h(P(x))) \geq p_1$
- if $Sim(q,x) \leq cS_0$ then $Pr_{\mathcal{H}}(h(Q(q)) = h(P(x))) \leq p_2$

When $Q(x) = P(x) = x$, we recover the vanilla LSH definition with $h(.)$ as the required hash function. Coming back to the problem of MIPS, if $Q$ and $P$ are different, the event $\{h(Q(x)) = h(P(x))\}$ will not have probability equal to 1 in general. Thus, $Q \neq P$ can counter the fact that self similarity is not highest with inner products. We just need the probability of the new collision event $\{h(Q(q)) = h(P(y))\}$ to satisfy the conditions in the definition of $c$-NN for $Sim(q,y) = q^T y$. Note that the query transformation $Q$ is only applied on the query and the pre-processing transformation $P$ is applied to $x \in S$ while creating hash tables. It is this asymmetry which will allow us to solve MIPS efficiently. In Section 3.3, we explicitly show a construction (and hence the existence) of asymmetric locality sensitive hash function for solving MIPS. The source of randomization $h$ for both $q$ and $x \in S$ is the same. Formally, it is not difficult to show a result analogous to Fact 1.

**Theorem 2** *Given a family of hash function $\mathcal{H}$ and the associated query and preprocessing transformations $P$ and $Q$, which is $(S_0, cS_0, p_1, p_2)$ -sensitive, one can construct a data structure for $c$-NN with $O(n^\rho \log n)$ query time and space $O(n^{1+\rho})$, where $\rho = \frac{\log p_1}{\log p_2}$.*

## 3.3 From MIPS to Near Neighbor Search (NNS)

Without loss of any generality, let $U < 1$ be a number such that $\|x_i\|_2 \leq U < 1$, $\forall x_i \in S$. If this is not the case then define a scaling transformation,

$$S(x) = \frac{U}{M} \times x; \qquad M = max_{x_i \in S} \|x_i\|_2; \tag{7}$$

Note that we are allowed one time preprocessing and asymmetry, $S$ is the part of asymmetric transformation. For simplicity of arguments, let us assume that $\|q\|_2 = 1$, the $\arg\max$ is anyway independent of the norm of the query. Later we show in Section 3.6 that it can be easily removed.

We are now ready to describe the key step in our algorithm. First, we define two vector transformations $P : \mathbb{R}^D \mapsto \mathbb{R}^{D+m}$ and $Q : \mathbb{R}^D \mapsto \mathbb{R}^{D+m}$ as follows:

$$P(x) = [x; \|x\|_2^2; \|x\|_2^4; ....; \|x\|_2^{2^m}]; \qquad Q(x) = [x; 1/2; 1/2; ....; 1/2], \qquad (8)$$

where $[;]$ is the concatenation. $P(x)$ appends $m$ scalers of the form $\|x\|_2^{2^i}$ at the end of the vector $x$, while Q(x) simply appends $m$ "1/2" to the end of the vector $x$. By observing that

$$Q(q)^T P(x_i) = q^T x_i + \frac{1}{2}(\|x_i\|_2^2 + \|x_i\|_2^4 + ... + \|x_i\|_2^{2^m}); \quad \|P(x_i)\|_2^2 = \|x_i\|_2^2 + \|x_i\|_2^4 + ... + \|x_i\|_2^{2^{m+1}}$$

we obtain the following key equality:

$$\|Q(q) - P(x_i)\|_2^2 = (1 + m/4) - 2q^T x_i + \|x_i\|_2^{2^{m+1}} \qquad (9)$$

Since $\|x_i\|_2 \leq U < 1$, $\|x_i\|^{2^{m+1}} \to 0$, at the tower rate (exponential to exponential). The term $(1 + m/4)$ is a fixed constant. As long as $m$ is not too small (e.g., $m \geq 3$ would suffice), we have

$$\arg\max_{x \in \mathcal{S}} q^T x \simeq \arg\min_{x \in \mathcal{S}} \|Q(q) - P(x)\|_2 \qquad (10)$$

This gives us the connection between solving un-normalized MIPS and approximate near neighbor search. Transformations $P$ and $Q$, when norms are less than 1, provide correction to the $L_2$ distance $\|Q(q) - P(x_i)\|_2$ making it rank correlate with the (un-normalized) inner product. This works only after shrinking the norms, as norms greater than 1 will instead blow the term $\|x_i\|_2^{2^{m+1}}$.

### 3.4    Fast Algorithms for MIPS

Eq. (10) shows that MIPS reduces to the standard approximate near neighbor search problem which can be efficiently solved. As the error term $\|x_i\|_2^{2^{m+1}} < U^{2^{m+1}}$ goes to zero at a tower rate, it quickly becomes negligible for any practical purposes. In fact, from theoretical perspective, since we are interested in guarantees for $c$-approximate solutions, this additional error can be absorbed in the approximation parameter $c$. Formally, we can state the following theorem.

**Theorem 3** *Given a $c$-approximate instance of MIPS, i.e., $Sim(q, x) = q^T x$, and a query $q$ such that $\|q\|_2 = 1$ along with a collection $\mathcal{S}$ having $\|x\|_2 \leq U < 1$ $\forall x \in \mathcal{S}$. Let $P$ and $Q$ be the vector transformations defined in (8). We have the following two conditions for hash function $h_{a,b}^{L2}$ (5)*
*1) if $q^T x \geq S_0$ then $Pr[h_{a,b}^{L2}(Q(q)) = h_{a,b}^{L2}(P(x))] \geq F_r\big(\sqrt{1 + m/4 - 2S_0 + U^{2^{m+1}}}\big)$*
*2) if $q^T x \leq cS_0$ then $Pr[h_{a,b}^{L2}(Q(q)) = h_{a,b}^{L2}(P(x))] \leq F_r\big(\sqrt{1 + m/4 - 2cS_0}\big)$*
*where the function $F_r$ is defined in (6).*

Thus, we have obtained $p_1 = F_r\big(\sqrt{(1 + m/4) - 2S_0 + U^{2^{m+1}}}\big)$ and $p_2 = F_r\big(\sqrt{(1 + m/4) - 2cS_0}\big)$. Applying Theorem 2, we can construct data structures with worst case $O(n^\rho \log n)$ query time guarantees for $c$-approximate MIPS, where

$$\rho = \frac{\log F_r\big(\sqrt{1 + m/4 - 2S_0 + U^{2^{m+1}}}\big)}{\log F_r\big(\sqrt{1 + m/4 - 2cS_0}\big)} \qquad (11)$$

We need $p_1 > p_2$ in order for $\rho < 1$. This requires us to have $-2S_0 + U^{2^{m+1}} < -2cS_0$, which boils down to the condition $c < 1 - \frac{U^{2^{m+1}}}{2S_0}$. Note that $\frac{U^{2^{m+1}}}{2S_0}$ can be made arbitrarily close to zero with the appropriate value of $m$. For any given $c < 1$, there always exist $U < 1$ and $m$ such that $\rho < 1$. This way, we obtain a sublinear query time algorithm for MIPS.

We also have one more parameter $r$ for the hash function $h_{a,b}$. Recall the definition of $F_r$ in Eq. (6): $F_r(d) = 1 - 2\Phi(-r/d) - \frac{2}{\sqrt{2\pi}(r/d)}\big(1 - e^{-(r/d)^2/2}\big)$. Thus, given a $c$-approximate MIPS instance, $\rho$

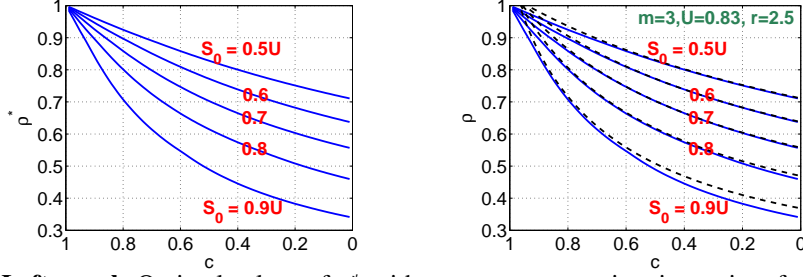

Figure 1: **Left panel**: Optimal values of $\rho^*$ with respect to approximation ratio $c$ for different $S_0$. The optimization of Eq. (14) was conducted by a grid search over parameters $r$, $U$ and $m$, given $S_0$ and $c$. **Right Panel**: $\rho$ values (dashed curves) for $m = 3$, $U = 0.83$ and $r = 2.5$. The solid curves are $\rho^*$ values. See more details about parameter recommendations in *arXiv:1405.5869*.

is a function of 3 parameters: $U, m, r$. The algorithm with the best query time chooses $U$, $m$ and $r$, which minimizes the value of $\rho$. For convenience, we define

$$\rho^* = \min_{U,m,r} \frac{\log F_r\left(\sqrt{1 + m/4 - 2S_0 + U^{2^{m+1}}}\right)}{\log F_r\left(\sqrt{1 + m/4 - 2cS_0}\right)} \quad s.t. \quad \frac{U^{2^{m+1}}}{2S_0} < 1 - c, \ m \in \mathbb{N}^+, \ 0 < U < 1. \quad (12)$$

See Figure 1 for the plots of $\rho^*$. With this best value of $\rho$, we can state our main result in Theorem 4.

**Theorem 4** *(**Approximate MIPS is Efficient**) For the problem of $c$-approximate MIPS with $\|q\|_2 = 1$, one can construct a data structure having $O(n^{\rho^*} \log n)$ query time and space $O(n^{1+\rho^*})$, where $\rho^* < 1$ is the solution to constraint optimization (14).*

### 3.5 Practical Recommendation of Parameters

Just like in the typical LSH framework, the value of $\rho^*$ in Theorem 4 depends on the $c$-approximate instance we aim to solve, which requires knowing the similarity threshold $S_0$ and the approximation ratio $c$. Since, $\|q\|_2 = 1$ and $\|x\|_2 \leq U < 1$, $\forall x \in \mathcal{S}$, we have $q^t x \leq U$. A reasonable choice of the threshold $S_0$ is to choose a high fraction of U, for example, $S_0 = 0.9U$ or $S_0 = 0.8U$.

The computation of $\rho^*$ and the optimal values of corresponding parameters can be conducted via a grid search over the possible values of $U$, $m$ and $r$. We compute $\rho^*$ in Figure 1 (Left Panel). For convenience, we recommend $m = 3$, $U = 0.83$, and $r = 2.5$. With this choice of the parameters, Figure 1 (Right Panel) shows that the $\rho$ values using these parameters are very close to $\rho^*$ values.

### 3.6 Removing the Condition $\|q\|_2 = 1$

Changing norms of the query does not affect the $\arg\max_{x \in \mathcal{C}} q^T x$. Thus in practice for retrieving top-ranked items, normalizing the query should not affect the performance. But for theoretical purposes, we want the runtime guarantee to be independent of $\|q\|_2$. We are interested in the $c$-approximate instance which being a threshold based approximation changes if the query is normalized.

Previously, transformations $P$ and $Q$ were precisely meant to remove the dependency on the norms of $x$. Realizing the fact that we are allowed asymmetry, we can use the same idea to get rid of the norm of $q$. Let $M$ be the upper bound on all the norms or the radius of the space as defined in Eq (7). Let the transformation $S : \mathbb{R}^D \rightarrow \mathbb{R}^D$ be the ones defined in Eq (7). Define asymmetric transformations $P' : \mathbb{R}^D \rightarrow \mathbb{R}^{D+2m}$ and $Q' : \mathbb{R}^D \rightarrow \mathbb{R}^{D+2m}$ as

$$P'(x) = [x; \|x\|_2^2; \|x\|_2^4; ....; \|x\|_2^{2^m}; 1/2; ...1/2]; \quad Q'(x) = [x; 1/2; ....; 1/2; \|x\|_2^2; \|x\|_2^4; ....; \|x\|_2^{2^m}],$$

Given the query $q$ and data point $x$, our new asymmetric transformations are $Q'(S(q))$ and $P'(S(x))$ respectively. We observe that

$$\|Q'(S(q)) - P'(S(x))\|_2^2 = \frac{m}{2} + \|S(x)\|_2^{2^{m+1}} + \|S(q)\|_2^{2^{m+1}} - 2q^t x \times \left(\frac{U^2}{M^2}\right) \quad (13)$$

Both $\|S(x)\|_2^{2^{m+1}}, \|S(q)\|_2^{2^{m+1}} \leq U^{2^{m+1}} \rightarrow 0$. Using exactly same arguments as before, we obtain

**Theorem 5** *(Unconditional Approximate MIPS is Efficient) For the problem of c-approximate MIPS in a bounded space, one can construct a data structure having $O(n^{\rho_u^*} \log n)$ query time and space $O(n^{1+\rho_u^*})$, where $\rho_u^* < 1$ is the solution to constraint optimization (14).*

$$\rho_u^* = \min_{0 < U < 1, m \in N, r} \frac{\log F_r\left(\sqrt{m/2 - 2S_0\left(\frac{U^2}{M^2}\right) + 2U^{2^{m+1}}}\right)}{\log F_r\left(\sqrt{m/2 - 2cS_0\left(\frac{U^2}{M^2}\right)}\right)} \quad s.t. \quad \frac{U^{(2^{m+1}-2)}M^2}{S_0} < 1 - c, \quad (14)$$

Again, for any $c$-approximate MIPS instance, with $S_0$ and $c$, we can always choose $m$ big enough such that $\rho_u^* < 1$. The theoretical guarantee only depends on the radius of the space $M$.

### 3.7 A Generic Recipe for Constructing Asymmetric LSHs

We are allowed any asymmetric transformation on $x$ and $q$. This gives us a lot of flexibility to construct ALSH for new similarities $\mathcal{S}$ that we are interested in. The generic idea is to take a particular similarity $Sim(x, q)$ for which we know an existing LSH or ALSH. Then we construct transformations $P$ and $Q$ such $Sim(P(x), Q(q))$ is monotonic in the similarity $\mathcal{S}$ that we are interested in. The other observation that makes it easier to construct $P$ and $Q$ is that LSH based guarantees are independent of dimensions, thus we can expand the dimensions like we did for $P$ and $Q$.

This paper focuses on using L2LSH to convert near neighbor search of $L_2$ distance into an ALSH (i.e., *L2-ALSH*) for MIPS. We can devise new ALSHs for MIPS using other similarities and hash functions. For instance, utilizing sign random projections (SRP), the known LSH for correlations, we can construct different $P$ and $Q$ leading to a better ALSH (i.e., *Sign-ALSH*) for MIPS [22]. We are aware another work [18] which performs very similarly to *Sign-ALSH*. Utilizing minwise hashing [2, 15], which is the LSH for resemblance and is known to outperform SRP in sparse data [23], we can construct an even better ALSH (i.e., *MinHash-ALSH*) for MIPS over binary data [21].

## 4 Evaluations

**Datasets.** We evaluate the proposed ALSH scheme for the MIPS problem on two popular collaborative filtering datasets on the task of item recommendations: (i) Movielens(10M), and (ii) Netflix. Each dataset forms a sparse **user-item matrix** $R$, where the value of $R(i, j)$ indicates the rating of user $i$ for movie $j$. Given the user-item ratings matrix $R$, we follow the standard PureSVD procedure [4] to generate user and item latent vectors. This procedure generates latent vectors $u_i$ for each user $i$ and vector $v_j$ for each item $j$, in some chosen fixed dimension $f$. The PureSVD method returns top-ranked items based on the inner products $u_i^T v_j$, $\forall j$. Despite its simplicity, PureSVD outperforms other popular recommendation algorithms [4]. Following [4], we use the same choices for the latent dimension $f$, i.e., $f = 150$ for Movielens and $f = 300$ for Netflix.

### 4.1 Ranking Experiment for Hash Code Quality Evaluations

We are interested in knowing, how the two hash functions correlate with the top-10 inner products. For this task, given a user $i$ and its corresponding user vector $u_i$, we compute the top-10 gold standard items based on the actual inner products $u_i^T v_j$, $\forall j$. We then compute $K$ different hash codes of the vector $u_i$ and all the item vectors $v_j$s. For every item $v_j$, we compute the number of times its hash values matches (or collides) with the hash values of query which is user $u_i$, i.e., we compute $Matches_j = \sum_{t=1}^{K} \mathbf{1}(h_t(u_i) = h_t(v_j))$, based on which we rank all the items.

Figure 2 reports the precision-recall curves in our ranking experiments for top-10 items, for comparing our proposed method with two baseline methods: the original L2LSH and the original sign random projections (SRP). These results confirm the substantial advantage of our proposed method.

### 4.2 LSH Bucketing Experiment

We implemented the standard $(K, L)$-parameterized (where $L$ is number of hash tables) bucketing algorithm [1] for retrieving top-50 items based on PureSVD procedure using the proposed ALSH hash function and the two baselines: SRP and L2LSH. We plot the recall vs the mean ratio of inner product required to achieve that recall. The ratio being computed relative to the number of inner products required in a brute force linear scan. In order to remove the effect of algorithm parameters $(K, L)$ on the evaluations, we report the result from the best performing $K$ and $L$ chosen from $K \in \{5, 6, ..., 30\}$ and $L \in \{1, 2, ..., 200\}$ for each query. We use $m = 3$, $U = 0.83$, and $r = 2.5$ for

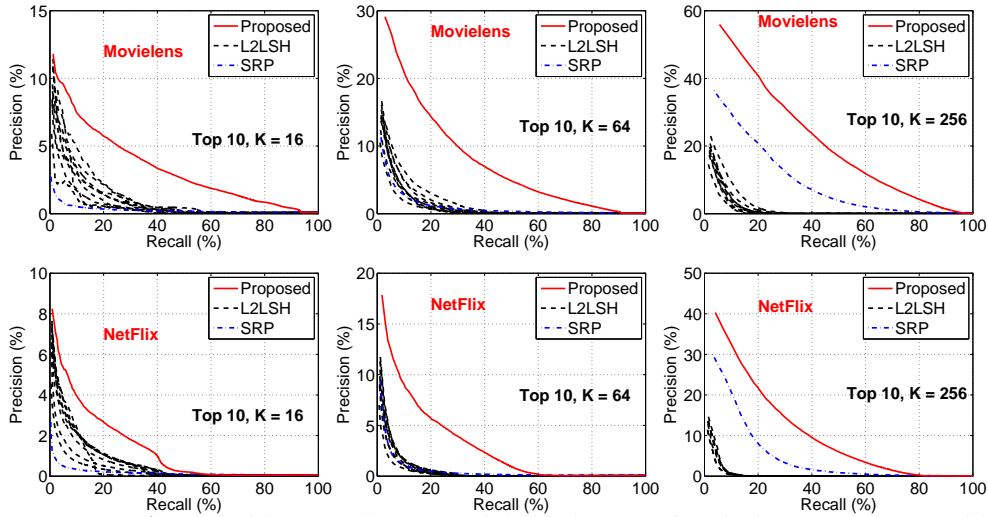

Figure 2: **Ranking.** Precision-Recall curves (higher is better), of retrieving top-10 items, with the number of hashes $K \in \{16, 64, 256\}$. The proposed algorithm (solid, red if color is available) significantly outperforms L2LSH. We fix the parameters $m = 3$, $U = 0.83$, and $r = 2.5$ for our proposed method and we present the results of L2LSH for all $r$ values in $\{1, 1.5, 2, 2.5, 3, 3.5, 4, 4.5, 5\}$.

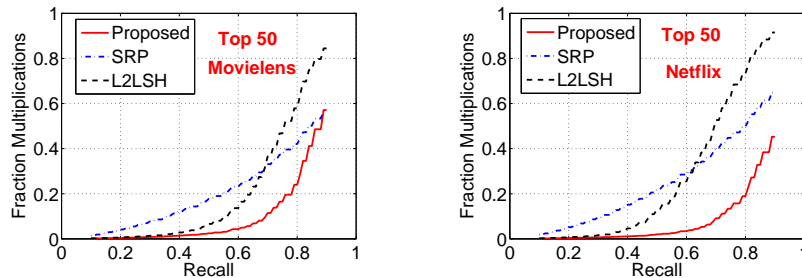

Figure 3: **Bucketing.** Mean number of inner products per query, relative to a linear scan, evaluated by different hashing schemes at different recall levels, for generating top-50 recommendations (Lower is better). The results corresponding to the best performing $K$ and $L$ (for a wide range of $K$ and $L$) at a given recall value, separately for all the three hashing schemes, are shown.

our hashing scheme. For L2LSH, we observe that using $r = 4$ usually performs well and so we show results for $r = 4$. The results are summarized in Figure 3, confirming that the proposed ALSH leads to significant savings compared to baseline hash functions.

## 5   Conclusion

MIPS (maximum inner product search) naturally arises in numerous practical scenarios, e.g., collaborative filtering. This problem is challenging and, prior to our work, there existed no provably sublinear time *hashing* algorithms for MIPS. Also, the existing framework of classical LSH (locality sensitive hashing) is not sufficient for solving MIPS. In this study, we develop *ALSH* (asymmetric LSH), which generalizes the existing LSH framework by applying (appropriately chosen) asymmetric transformations to the input query vector and the data vectors in the repository. We present an implementation of ALSH by proposing a novel transformation which converts the original inner products into $L_2$ distances in the transformed space. We demonstrate, both theoretically and empirically, that this implementation of ALSH provides provably efficient as well as practical solution to MIPS. Other explicit constructions of ALSH, for example, ALSH through cosine similarity, or ALSH through resemblance (for binary data), will be presented in followup technical reports.

## Acknowledgments

The research is partially supported by NSF-DMS-1444124, NSF-III-1360971, NSF-Bigdata-1419210, ONR-N00014-13-1-0764, and AFOSR-FA9550-13-1-0137. We appreciate the constructive comments from the program committees of KDD 2014 and NIPS 2014. Shrivastava would also like to thank Thorsten Joachims and the Class of CS6784 (Spring 2014) for valuable feedbacks.

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
