[Reviews · NeurIPS 2014]

Submitted by Assigned_Reviewer_6

This paper generalizes the LSH method to account for the (bounded) lengths of the data base vectors, so that the LSH tricks for fast approximate nearest neighbor search can exploit the well-known relation between Euclidian distance and dot product similarity (e.g. as in equation 2) and support MIPS search as well. They give 3 motivating examples where solving MIPS vs kNN per se is more appropriate and needed. Their algorithm is essentially equation 9 (using equation 7 compute vector reformulations Q(q) and P(x) of the query a database element respectively). This is based on apparently novel observation (equation 8) that the distance from the query converges to the dot product plus a constant, when a parameter m which exponentiated the P(x) vector elements is sufficiently large (e.g. just 3 is claimed to suffice, leading to vectors Q(q) and P(x) which are just that m times larger than the original input dimensionality.

The work seems novel and potentially significant. One concern though is that the experimental results actually provided do not clearly demonstrate an example where this method is better in practice than brute force linear scan. The authors claim their "scheme thus offers around 5x speedup over linear scan", at the end of Section 4.2, but they do not clearly state how they implemented linear scan or how much wall clock time linear scan took. The reason this is a concern is that linear scan can be implemented on modern hardware (multicore CPUs or GPUs) to be ~10x faster using modern libraries (e.g. BLAS SGEMM matrix multiply operatons, using batches of queries and the fully batch of database vectors) than the vector based, one-query-at-a-time loops that presumably the proposed hased-baed scheme requires, due to the cache-awareness of tile-based SGEMM ops for modern hardware. Furthermore, hashing schemes will have additional overheads that must be demonstrated to not add significance constants -- especially when the speedup versus simple, low-constant-overhead linear scan approaches are relatively low (such as 5x here). In short, this paper's focus on the 5x less dot products being performed for 90% accuracy does not prove that the method actually is, or will be for practical problems in general, faster in a practical wall clock sense than well-implemented linear scans.
Summary: Proposed an apparently novel and interesting approach to reformulate vectors to enable LSH methods to apply to MIPS problems. The experimental results are weak in terms of establishing the actual practicality (e.g. in a wall clock sense) versus linear scan; the paper would be much more impactful and exciting if this was fixed via better experiments and a clearly fair state-of-art implementation of SGEMM-based linear scan, which should be feasible for the authors to add.

Submitted by Assigned_Reviewer_13

This paper shows how to use LSH for the task of finding points with max inner product with a query point. There are two major ideas: (i) using different hash functions for query and database points, and (ii) extending each data point with dimensions that capture its norm (line 225). The ideas can be called simple or elegant, depending on point of view. The paper is well-argued and persuasive, and experimental results are good. People will use this method in practice, or else some future improvement of it.

DETAILED COMMENTS

026: "outperforms" is too vague. Clarify: accuracy, wall clock time, number of dot products, and/or what?

060: This paragraph is correct, but it raises a question. In standard CF, dot products u_i v_j are trained to estimate ratings which are bounded (e.g. 1 to 5) or to estimate a binary flag (viewed or not). So the dynamic range of u_i v_j is restricted. Do norm(v_j) magnitudes really vary a lot?

063: Brute force computation of dot products can be done at maximum theoretical gigaflops on CPUs and GPUs. Any data structure that does 10% of brute force calculations, but that uses complex logic to decide *which* dot products, is likely to be slower in practice than a vectorized brute force implementation.

070: This max scoring [ref. 6] is similar to max pooling, which is one of the major new ideas in deep nets for vision.

079: The same argument applies to linear multilabel learning, which is very important in practice.

387: Show similar plots for wall clock time. See comment under 063.

WRITING

Organization and English are overall good. The argument flows well and the paper is readable. A few low-level mistakes:
172 and elsewhere: "can not" should be "cannot".
173 and 371: Commas are incorrect.
194: should be "losing"
227: should be "scalars"

COMMENTS ON THE AUTHORS' RESPONSE

Please find a way to include all your general points about hashing in the paper. If necessary, shorten somer math or experiments and refer to an online long version of the paper. These observations will be useful to practitioners and to other researchers. Many people do not understand or appreciate them fully.

Summary: Good, well-argued paper with useful results.

Submitted by Assigned_Reviewer_23

The paper addresses the problem of fast nearest-neighbor search over dot product similarities (aka MIPS). Theoretical analysis is provided explaining why standard LSH techniques are insufficient, a novel algorithm is derived, and extensive experimental evaluation is conducted.

The paper is based on a very neat, out-of-the-box idea: decoupling the preprocessing and query-time hash functions, which allows obtaining an exciting new algorithm. Authors should be commended for a derivation is very easy to follow, from the negative result derivation for classic LSH to coming up with the relationships between \rho and the rest of the hashing parameters.

The result is very exciting, as it solves an important open problem, creating a method for fast hashing-based retrieval for an important class of similarities, that previously could only approximate dot product via cosine distances via Charikar's method. The detailed analysis showing viability and practical recipes for parameter settings on p.6 is excellent, and provides a ready-to-use recipe to practitioners.

The paper should also be commended for detailed experimental evaluation. It sets the correct baselines (gold-standard retrieval based on SVD for collaborative filtering), and compares to the obvious alternatives (Charikar's method and L2LSH). The results clearly demonstrate the benefits of the approach, which a nice motivational application behind it.
Summary: Excellent paper solving an important open problem via a creative *and* principled algorithm.
Author Feedback
Author rebuttal: We would like to thank all Reviewers for their encouraging comments and investing time to improve the research output of our hard work. We are very glad to know that Reviewers found our work novel and exciting.

Reviewers' comments will be carefully addressed in the revision. Here, we would like to focus on the interesting issue about brute-force implementation of linear search with GPUs, raised mainly by Reviewer_6.

Reviewer_6 brought up an interesting (and very important) general discussion on the practicality of whole hashing framework (for near neighbors or inner products) in the era where linear scan and brute force computation can be made faster with modern GPU based implementations.

The hashing framework offers many advantages over fast brute force implementation which is why it is still a favorite in practice for search. We provide these arguments as the part of general discussion, which is out of the scope of this paper.

1) While GPUs are great. (In fact hashing should make GPU implementations even more efficient and practical, more on this later...) In general, GPU implementations are not energy efficient since they are doing all the computations for every query, whereas, hashing scheme needs very less computation. Energy efficiency is becoming a big concern these days for sustainability.

2) In many practical scenarios, we don’t even need to evaluate inner products at all for getting an estimate of rankings. For instance, in reference [6] (and plenty others application papers), to estimate the rank of x, they only use the number of times a given point x occurs in the bucket of query q. The monotonicity of collision itself gives a crude approximation of ranking which was used in the paper to compute a set of near neighbor (by simple thresholding) and it seems to work reasonably. (No scanning no distance computation).

3) Hashes itself can be used to estimate similarities very cheaply and accurately. We don’t need the actual data points which may be problematic to store, so less overhead on memory. (monotonicity of collision is enough to estimate, as monotonic functions are invertible)

4) In the era, where data sharding (distribute data) across the globe is common for web industry, hashing is a boon. It is massively parallelizable. Hash tables can be created locally (local data stays local), all we need is communication of few hashes of the query to the appropriate node and the final aggregation (reducer) is only finding the global max from the local max.

5) The amount of memory in each GPU is quite limited, so we cannot process all the data at once and there is a needs to scan and operate on chunks (for every query).

Hashing schemes offers significant practical impact. Having hashing scheme for inner products which is frequently used is many applications is of special interest which was thought to be hard with existing LSH framework. We found that the famous LSH framework is fixable (for inner products) if we introduce asymmetry, which is exciting. We do believe that our work will have significant impact both in theory as well as practice and will lead to more interesting asymmetric LSH for many new similarities, which were not possible before.

We once again thank all the reviewers for their time and efforts.